# Sustainable Feedstocks and Challenges in Biodiesel Production: An Advanced Bibliometric Analysis

**DOI:** 10.3390/bioengineering9100539

**Published:** 2022-10-10

**Authors:** Misael B. Sales, Pedro T. Borges, Manoel Nazareno Ribeiro Filho, Lizandra Régia Miranda da Silva, Alyne P. Castro, Ada Amelia Sanders Lopes, Rita Karolinny Chaves de Lima, Maria Alexsandra de Sousa Rios, José C. S. dos Santos

**Affiliations:** 1Instituto de Engenharias e Desenvolvimento Sustentável, Universidade da Integração Internacional da Lusofonia Afro-Brasileira, Campus das Auroras, Redenção 62790970, CE, Brazil; 2Departamento de Engenharia Mecânica, Grupo de Inovações Tecnológicas e Especialidades Químicas—GRINTEQUI, Universidade Federal do Ceará, Bloco 715, Campus do Pici, Fortaleza 60440554, CE, Brazil; 3Departamento de Engenharia Química, Universidade Federal do Ceará, Campus do Pici, Bloco 709, Fortaleza 60455760, CE, Brazil

**Keywords:** biodiesel, biofuel, feedstocks, sustainability, research articles, bibliometric analysis

## Abstract

Biodiesel can be produced from vegetable oils, animal fats, frying oils, and from microorganism-synthesized oils. These sources render biodiesel an easily biodegradable fuel. The aim of this work was to perform an advanced bibliometric analysis of primary studies relating to biodiesel production worldwide by identifying the key countries and regions that have shown a strong engagement in this area, and by understanding the dynamics of their collaboration and research outputs. Additionally, an assessment of the main primary feedstocks employed in this research was carried out, along with an analysis of the current and future trends that are expected to define new paths and methodologies to be used in the manufacture of biodegradable and renewable fuels. A total of 4586 academic outputs were selected, including peer-reviewed research articles, conference papers, and literature reviews related to biodiesel production, in the time period spanning from 2010 to 2021. Articles that focused on feedstocks for the production of biodiesel were also included, with a search that returned 330 papers. Lastly, 60 articles relating to biodiesel production via sewage were specifically included to allow for an analysis of this source as a promising feedstock in the future of the biofuel market. Via the geocoding and the document analyses performed, we concluded that China, Malaysia, and India are the largest writers of articles in this area, revealing a great interest in biofuels in Asia. Additionally, it was noted that environmental concerns have caused authors to conduct research on feedstocks that can address the sustainability challenges in the production of biodiesel.

## 1. Introduction

The process of deescalating the use of conventional fuels faces several hurdles worldwide, and despite the growing scientific evidence of the viability of other biofuel production routes, governmental and regulatory limitations accentuate these difficulties and forestalls further progress [1,2,3,4,5,6,7]. Despite these hindrances, biodiesel production shows a rapid growth globally owing to the security offered by this type of fuel and its considerably smaller environmental footprint [8,9,10,11,12,13]. There is also an increased pressure on governments to adopt and implement more sustainable production processes. Thus, the demand for biofuel consumption is predicted to grow significantly in the coming years due to governmental policies in some countries pushing for a shift towards the use of renewable energy, to the rising prices of petroleum-based fuels, and to the emerging concerns related to pollutants [14].

Bibliometric analyses are a well-known and rigorous method for the analysis of large amounts of scientific information [15]. It allows for the visualization of the development of a specific research topic, while also revealing important information about the areas that are rapidly evolving in that field [16,17]. The data sets encompassing the structure of a bibliometric analysis are usually extremely large, in the scale of hundreds or even thousands, and they can describe specific research numbers, such as the volume of citations and/or publications and the number of occurrences of certain keywords, among others. However, the interpretations gathered and discussed are not only objective in nature, as in performance analyses, but they can also comprise more subjective considerations, such as topical analyses [18].

This article aimed to critically evaluate, from a qualitative and quantitative standpoint, academic papers published on themes related to the specific aforementioned area via the creation of a highly specific database. Articles in this database included those related to biodiesel production and those that were published between 2010 and 2021. Quantitative and qualitative conditions were set and entered within the search tool available on the Web of Science Core Collection website (https://www-webofscience.ez373.periodicos.capes.gov.br/wos/woscc/basic-search, accessed on 8 August 2022). The articles selected were specifically related to feedstocks employed in the manufacture of biodiesel (e.g., animal oils, vegetable oils, oils from microorganisms, etc.).

In addition to the considerations above, the relevance of sewage sludge as a specific feedstock for biofuel production was analyzed, due to its strong future potential in this specific market. Sewage sludge has been recognized worldwide as being a strong lipid feedstock for biofuel production due to its wide availability and good concentration of lipids. In the US alone, around 6.2 million dry metric tons of sludge are produced annually by sewage treatment plants, and this number is expected to increase in the future due to the ever-growing urbanization and industrialization observed in most countries [19]. This raw material became attractive in the biofuel market in mid-2010, mainly owing to its increased production at that time. In the European Union alone, sludge production is forecasted to have reached the 13-million-ton mark in 2020 [20].

The main aim of this article was to showcase the evolution of the literature related to the research on biofuel production worldwide via an analysis process that facilitates the visualization of future trends in related emerging studies. Thus, the raw materials that presented the highest relevance in the database built were assessed, along with the frequency of studies citing these sources. Given the accumulation of a great number of studies across the years, we had to set boundaries concerning search criteria and targeted conditions, as described in Section 2.

As presented in Figure 1, the growth of publications on the manufacture of biodiesel is noticeable. It is also noteworthy that, especially in the year 2021, the number of publications related to feedstocks for biodiesel production was considerably higher than in previous years, with about 9.8% of the total number of publications coming from that year (44 documents). The second most relevant year is 2018, with 8.9% of the total publications (41 documents), followed by 2017, with approximately 7.2% (32 documents). Regarding articles on sewage sludge, 2016 saw the greatest number of publications (with a total of 14).

## 2. Materials and Methods

### 2.1. Data Sources

Some search parameters were previously set on the Web of Science Core Collection website when using programs to collect the data to compose our advanced bibliometric analysis. The database files were accessed and downloaded via the login credentials provided by the CAPES PERIODICOS platform (https://www-periodicos-capes-gov-br.ezl.periodicos.capes.gov.br/index.php, accessed on 9 August 2022).

As shown in Figure 2, first, the term “Biodiesel Production” was entered in the search bar, along with the filter “Title”, into their central database, which contained all the subsequent databases. Then, a new row was added and the term “2010–2021” was entered within the filter “Year Published”. In total, 4917 articles fitting these search parameters were returned, but to ensure a more refined analysis, we proceeded to discard the publications that showed little relevance to this bibliometric analysis. Refinements were made for “Document Types” (limiting the search to articles, conference papers, and review articles only) and Language (English). The new, targeted search returned 4586 articles.

In addition, two further databases were generated from the main one above in order to allow us to delve deeper into other subtopics of interest. The first database was related to feedstocks for biodiesel production, where a third search line was added with the term “feedstocks” in the “All Fields” section and with the same previous refinements. Overall, 330 articles published between 2010 and 2021 matched these criteria. The second derived database was related to biodiesel production using sewage sludge, with a third row being added with these terms. A total of 60 related articles were then identified.

The methodology described and used in this research was chosen based on the literature evidenced by the article *A bibliometric analysis of sustainable oil and gas production research using VOSviewer* published in 2022 [21]. The authors reinforce that such a strategy facilitates the construction of a database that is consistent with the aim being proposed and that improves the visualization and exploration of results in this area of research. The three databases obtained in our work were all crucial for this analysis, with the largest of them providing the comparison basis that created, with the delimitation and search conditions shown in Figure 2, the second and third databases, which returned 330 and 60 documents, respectively. As the central objective of this work was to explore mainly the second database, which deals with raw materials, the third could then serve as an additional dataset to explore the importance of the topic it covers in the advancement of this research topic. Despite the limitations observed, another differentiating characteristic of this work was the inclusion of articles that provide important information, but that are not necessarily a part of the repertoire usually treated in the programs used to process these data for analysis. The literature on biodiesel is characterized by a wide variety of works scattered through various areas of study, and those were used to enrich this work.

Figure 2 above is a categorical representation of the search process followed for building the databases used in this work. Each block shows one step taken in our research, from entering the key terms and constraints used in the Web of Science search engine, to the analysis methodology to be followed. Such visual representations serve as more simplified demonstrations, aiming to describe how the research was carried out, since they do not contain any specific descriptions that can only be understood through reading the main body of text.

### 2.2. Data Analysis

For the bibliometric analysis, three programs were used: CiteSpace (v.5.8.R3 Philadelphia, PA, USA), CiteSpace was developed by Dr. Chaomei Chen from Drexel University, USA. VOSviewer (version 1.6.17 Leiden, South Holland, The Netherlands), VOSviewer software, developed at Leiden University’s Centre for Science and Technology Studies (CWTS), Leiden University, the Netherlands. ArcGIS (version 10.5 Redlands, California, USA), ArcGIS is a product suite developed by Environmental Systems Research Institute (Esri), Redlands, California, United States. ArcGIS 10.5 was used to analyze the geographical distribution of publications, while VOSviewer was used for data visualization [22]. CiteSpace identified and predicted the future possible research subareas in this area by using keywords and clusters [23]. 

## 3. Results and Discussion

### 3.1. Bibliometric Analysis

#### 3.1.1. Publications: General Results

The initial search on the Web of Science Database Core Collection website resulted in 4586 research outputs from 2010 to 2021, the most relevant of which was published in March 2012. The title of this specific article is *Biodiesel production from microalgae*, in which the use of microalgae in the manufacture of different energy sources, such as oil and biofuels, including biodiesel, were discussed [24]. Furthermore, the authors presented the various factors of relevance in the area of microalgae valorization for biodiesel, microalgae cultivation, lipid extraction, and transesterification reactions, among other aspects. By analyzing all the articles in the database, it was possible to observe that, among the raw materials commonly mentioned, microalgae were the most widely employed, being reported in 40 articles. This is higher than the number of mentions of soybean oil, a raw material with a relatively high rate of employment in the production of biofuels.

Microalgae, however, as well as other feedstocks, still face a cost–benefit challenge. Nevertheless, great advances are expected to be achieved in the areas of cultivation, harvesting, lipid extraction, transesterification, and biomass-processing technologies that could render biodiesel production from microalgae more viable and more easily commercialized in the near future [25]. Optimizing the manufacture of biofuels is a paramount step towards market acceptance and implementation, as this directly influences the economics and yields of the final product. The optimization of the variables and conditions in these processes are also of high relevance for the entire production process [26]. Additionally, the studies carried out on the topics above have been successful in demonstrating how key environmental concerns have been addressed through the steps that have been taken to make the biofuel market more sustainable. This trend is confirmed by the prominence of related articles present in the literature repertoire used in this research, mainly when considering the number of citations and the relevance of the authors of the papers. These factors are described in subsequent sections of this article.

#### 3.1.2. Distribution of Scientific Journals

From the reference database created, we observed that the manuscripts relating to raw materials for biodiesel were published across 154 journals, with an average of approximately 2.1 publications per journal. This indicates that a great interest in this research theme is shared by several areas, but the number of relevant publications can still be considered low. Judging by the vast number of journals analyzed, it is possible to note that distinct groups of researchers have explored this topic via different methodologies of scientific analysis.

Table 1 shows the list of the ten most relevant scientific journals in the biodiesel research area, classified by number of citations. Together, these ten journals are responsible for more than a quarter of the total number of the publications analyzed. The journal *Renewable & Sustainable Energy Reviews*, which ranks first on this list, has an impact factor of 14.982 and a total of 33 articles published in the area. These account for 9.9% of the total number of documents, and the articles published in this periodical have amassed 8250 citations over the years. The second journal on the list is *Bioresource Technology*, with an impact factor of 9.642 and 16 articles published (4.8% of the total documents) that gathered 1785 citations. The ranking list is mostly composed of European journals, with *Renewable & Sustainable Energy Reviews* being the only American journal. Nevertheless, it has the highest impact factor of all the journals listed and the highest number of citations and publications, with an average of 250 citations per article. This shows a high concentration of articles in the area of feedstocks for biodiesel production in a single scientific journal.

#### 3.1.3. Distribution by Country and Institution

Analyses made from the country information declared by the authors across the articles in our database allowed us to verify that the ten countries that published most articles on biodiesel feedstocks contributed 64.5% of the total amount of publications from the 56 countries identified (Table 2).

It is noteworthy that, although there is a great demand and interest for this research area, the resulting publications are concentrated within just a few regions of the world. China is responsible for 12.8% of the published articles (53 articles), thus occupying first place on the list. It is followed by Malaysia, with 11.6% (48 articles), and India, with 11.6% (48 articles).

To evaluate the relevance and impact of the published articles, it is necessary to also consider the number of citations. Malaysia has the highest number of citations (3715) across its 48 published papers, followed by Portugal, with 10 published papers that have amassed 3323 citations, and China, with 53 published papers and 2555 citations. Figure 3 shows the distribution of articles by country and specifies the 27 geographical regions that published at least five articles on the topic.

Figure 4 illustrates publication collaborations among countries, within the same constraints as above regarding research area and time period. China and the United States, for example, present a strong collaborative relationship between authors despite being located in different continents. This can be explained by the notable leadership of these two countries concerning world economic power, so there is a natural interest in forming scientific partnerships to address shared environmental concerns in the search for sustainable products. This can be observed in Figure 1, where there is a clear growing interest in studies related to feedstock for biodiesel production in both these countries. The most significant collaborative relationship, however, is between Malaysia and Indonesia. This is most likely due to their geographical proximity in Southeast Asia, which demonstrates the similarity between their economic and socio-environmental interests towards the production of biofuels.

It can also be noted that the articles published were the result of the work of 481 organizations across 56 different countries, which also confirms the high interest in research in this area. Nevertheless, the number of academic outputs is concentrated primarily in only ten countries, which produced two-thirds of the total number of documents. Among them are China, India, the United States, and Brazil, which indicates that there is a strong interest for this area especially in countries with large economies and a notable involvement in socio-environmental policies.

Another point worthy of note was that more than 50% of the organizations identified had only one publication in the field. In addition, only a small number of organizations were perceived to conduct consistent research and constantly publish in the area. Among the identified papers, there is a peculiarity related to the Polytechnic Institute of Porto (IPP). This organization has published only one article, *Microalgae for biodiesel production and other applications: A review* [27], but this output alone has gathered 3160 citations, the highest number of citations among all documents. This shows that the importance of each article is dictated by the high quality of the scientific research performed, thus some of them receive a more significant number of citations than others. The University of Agriculture in Faisalabad (UAF) can also be highlighted with three publications [28,29,30] that obtained only seven citations (an average of 2.3 citations per document), also confirming the above trend. Naturally, there may be other factors that cause this discrepancy, such as the point in time when the article was published, among others.

It is clear that many of the collaboration links are geographically related. However, there are some countries that have overcome the regional barrier and sought extracontinental academic cooperation, which is the case of (and between) the United States, China, and Malaysia. Brazil, for example, is a country that cooperates strongly with European countries such as Portugal and Spain. This is likely to happen due to their linguistic proximities and the fact that the country has a great relevance for world agribusiness, apart from showing high interest in using sustainable and renewable raw materials for the manufacture of various chemicals [31].

A network map representation was built to enable a better visualization of the links between organizations, and it is shown in Figure 5. To build the map, a minimum requirement of 250 citations per institution was set for the period analyzed, which identified 27 institutions from the database. The most relevant of them include the University of Porto, the Polytechnic Institute of Porto (IPP), and the University of Malaya.

In addition to the analyses performed by studying the above clusters, geocoding was also used by transforming locations into points on the map. This allowed for a deeper observation of the geographical reference of the institutions identified in the collated database. Figure 6 illustrates the representation of the geocoded locations, which reveals a large concentration of these organizations in North America, Europe, and South Asia. The United States and India, which are countries of continental proportions, clearly stand out when exploring the data from this perspective. Furthermore, Figure 5 shows that, of the most relevant institutions, only a few have a co-authorship relationship among the analyzed articles, showing that there may be a sense of individualism in the scientific groups highlighted in this study. From the 27 institutions that published papers containing at least 250 cumulative citations, only 11 (40.7%) showed collaboration with other organizations.

A total of 1304 different authors were identified across the 332 documents in the database, indicating a wide range of scientific collaboration among the researchers with the most significant influence in this area. Figure 7, which was built by selecting authors with 450+ citations, depicts these collaborative relationships. The article with the highest number of citations is entitled *Microalgae for biodiesel production and other applications: A review* [27], with 3160 citations. However, it can be noted that its author, Mata, had little collaborative input in other works. On the other hand, the author Mahlia collaborated in nine highly cited documents, including *Patent landscape review on biodiesel production: Technology updates* [32], published in 2020 and presenting 152 citations as of the writing of this review. Of the 977 authors with more than five citations, only two had no co-authorship relationship, i.e., were found isolated in two different clusters, which represented 1.07% of the collaboration sets. It can be concluded that most of the research in this specific area is achieved through collaborative clusters, highlighting a high appreciation for cooperative research despite the geographical dispersion of researchers.

#### 3.1.4. Most Cited Articles

The ten most cited articles on raw materials for biodiesel production have gained a collective number of 7219 citations, with the document occupying first place in this ranking amassing 3160 (43.7%) of these. As mentioned earlier, Mata’s work discussed the current status of biodiesel production by microalgae, focusing on the cultivation, harvesting, and processing techniques of the main microalgae species, along with the advantages and disadvantages of their use [27]. This article is of paramount relevance for the literature on this specific feedstock, given its use in other articles that compose the list presented in Table 3. Apart from the first position, microalgae are also mentioned in the titles of the articles in positions three, four, and nine. However, we can also note that outputs using these materials are now considerably old (2010–2012), which may hint at a changing scenario and a newfound appreciation of other raw materials that may currently be more advantageous or relevant.

The second most cited article on the list is *Non-edible vegetable oils: A critical evaluation of oil extraction, fatty acid compositions, biodiesel production, characteristics, engine performance, and emissions production* [33], which was cited by 645 papers within the period analyzed. This article discusses biodiesel production via non-edible vegetable oils, reviews recent related studies, and analyzes general aspects, advantages, and disadvantages of the commodities regarded as second-generation feedstocks. Among them are *Jatropha curcas, Pongamia pinnata* (Karanja), *Moringa oleifera*, and *Aleurites moluccanas*.

The visualization and analysis done on the keywords of the articles from our database, illustrated in Figure 8, revealed that, during the period spanning from January 2020 to December 2021, cooking oil residues showed the highest relevance (29 keyword occurrences), which may indicate a strong interest in this material, considering that it stands out even among other key words that are not necessarily raw materials. Palm oil and soy oil also appear in this list, with nine and six occurrences, respectively.

It is important to highlight that the journals that occupy the first and second positions in the ranking shown in Table 1 (Renewable & Sustainable Energy Reviews and Biotechnology Advances) are responsible for 8 of the 10 most cited in the area of feedstocks for biodiesel production (Table 3). This indicates that there is a large concentration of studies carried out by specific scientific groups. In addition, they constitute almost 15% of all publications in the collated database, among the 154 journals analyzed. Regarding the number of citations, the first journal earned 8257 (43% of the total), and the second, 1790 (9.3% of the total). This points to the high relevance of the publications in these journals and the wide dispersion of studies with less prominence across a great array of periodicals.

The article entitled Patent landscape review on biodiesel production: Technology updates [32], despite its smaller number of citations comparatively to the articles in Table 3, shows great relevance owing to its very recent publication date (February 2020), having already gathered 152 citations. It appears to be more relevant, for example, than the Fractional characterization of jatropha, neem, moringa, trisperma, castor and candlenut seeds as potential feedstocks for biodiesel production in Cuba [42], which is almost 10 years older and, comparatively, only received 114 citations. Mahlia is also a co-author in other articles, as shown in Figure 7 (this author is in a prominent cluster), which attests to the relevance of their research methodology. The article in question is a review of 1660 patents related to biodiesel production. The paper organized the patents into five categories: raw materials, pretreatment methods, catalysts, reactors, and processing methodologies.

#### 3.1.5. Research Areas

From the database compiled, we could observe that the 330 selected documents belonged to 25 research areas related to feedstocks for biodiesel production, from 2010 to 2021. Figure 9 shows that the most prominent them of was energy fuel, as 26.4% (163 out of 618) of the number of occurrences were from this area. Engineering was the second largest research area, with 16% of the total number of entries (99 occurrences). However, it can be seen that, despite the large number of occurrences in the area of energy fuels, there is not a high concentration in one single research area. The percentages across areas are similar, and the number of occurrences is fairly well distributed. The prominent relationship between energy fuels and engineering reveals that the market seeks scientific research that creates innovative solutions for solving existing problems related to the raw materials used for biodiesel production.

When analyzing the database keywords, which totaled 1536 occurrences, it was possible to observe that the most used terms were “biodiesel” (225 occurrences), followed by “transesterification” (145 occurrences), and “waste cooking oil” (91 occurrences). This shows that this raw material is the most significant in the studies related to biodiesel production, given its prominence in the documents. In Figure 8, the keyword analysis was limited to the years 2020 and 2021, in order to enable the observation of the direction of the scientific research and the emerging feedstock trends for the future of biodiesel production.

There seem to not yet exist highly relevant studies on feedstocks that have great potential for the future of the market, as is the case of sewage sludge. From the search carried out on the Web of Science Core Collection site, there are only 60 documents that mention this material. The article with the most relevance is entitled Recent development on sustainable biodiesel production using sewage sludge [43], which gives an overview of biodiesel production via the use of municipal sewage sludge as a highly available and economical waste, since it is produced on a large scale due to increased urbanization worldwide. It also has zero added value. This article only has 13 citations, suggesting that the raw material is still underexplored, in comparison to soybean and waste cooking oil, for example.

It is essential to point out that a refinement was made in the database search by discarding the works that are not of interest for the visualization of trends in the proposed theme. Therefore, the limitation of inclusion to only research articles, conference papers, and review articles enabled us to obtain more precise and relevant conclusions. Furthermore, the search for documents related to feedstocks in general and for sewage sludge as a raw material made it necessary for us to search for these terms specifically, i.e., the terms “Sewage Sludge” and “Feedstock” were entered in the search field “All fields”.

### 3.2. Feedstocks for Biodiesel Production

#### 3.2.1. Classification of Feedstocks

The feedstocks used for biodiesel production are varied and can be classified into the four following categories: edible oils, inedible oils (e.g., inedible vegetable oils, residual oils), oils from microorganisms, and animal fats (Figure 10). The figure shows some of the feedstocks most used in the production process of biofuel, along with their classification. It is necessary to highlight the importance of those that are not edible, given that the biodiesel market does not intend to compete with the food industry. Edible oils are highly valued, which leads to price competition and causes these products to become more expensive [44].

The rapid growth of the world population and the extensive human consumption of edible oils can cause serious problems, such as starvation, in developing countries. Therefore, a category that becomes very promising as an alternative material for biodiesel production is non-edible oils. Due to the high demand for edible oils in the food industry, their prices are higher than fossil fuels, and this fact alone highlights the importance of the lower cost of cultivation of non-edible vegetable oils [45]. Oils classified as non-edible can be obtained from native plants distributed in different geographical areas of the globe. Some examples are babassu (*Orbignya* sp.), mahua (*Madhuca indica*), castor oil plant (*Ricinus communis*), jatropha (*Jatropha curcas* L.), macaúba (*Acrocomia aculeata*), andiroba (*Carapa guianensis*), karanja (*Pongamia pinnata*), seed oil from *Pistacia chinensis bge* and crambe (*Crambe abyssinica Hochst*), pinnai (*Calophyllum inophyllum*), rubber tree seed (*Hevea brasiliensis*), and coconut (*Cocos nucifera*) oil [31].

Among the raw materials reported, microalgae and waste cooking oil are highly relevant. Among the 332 publications in the database, 20 and 19 publications present the terms “waste cooking oil” or “microalgae” in their title, respectively. Together, they represent 11.7% of the documents. Accordingly, it is important to highlight that the evaluation criteria in this case maintained the same refinements of the initial search, but with the added logical operator “AND” with the following text: “waste cooking oil” or “microalgae”. The articles reporting research that employ microalgae as the main object of study can be seen in Table 3, occupying prominent positions. The article with the highest number of citations that uses waste cooking oil as raw material is entitled A magnetically separable SO_4_/Fe-Al-TiO(2) solid acid catalyst for biodiesel production from waste cooking oil [46]. It presents a new magnetic solid acid catalyst that was synthesized and used in the production of biodiesel through the transesterification of waste raw material. This scientific output merited 115 citations during the analyzed period.

Performing the same analysis on sewage sludge, it was possible to identify only two documents that present this raw material in their title—Biodiesel Production from Sewage Sludge: New Paradigm for Mining Energy from Municipal Hazardous Material [47], with 83 citations, and Efficient extraction of lipids from primary sewage sludge using ionic liquids for biodiesel production [48], with 25 citations. The first article discusses the economic feasibility of biodiesel production using the lipids extracted from sewage sludge owing to the high oil yields and the lower value of this feedstock compared to those conventionally used. The second describes a new lipid extraction method from wet primary sludge for biodiesel production with ionic liquids.

#### 3.2.2. Sewage Sludge

Studies involving biodiesel production through sewage sludge are scarce due to there being currently little interest in this research. However, according to the publications collected for this analysis, there are many advantages to using this raw material for manufacturing biofuels. Feedstocks that fall into the edible group (such as sunflower oil, rapeseed oil, and soybean) show low cultivation costs and land use requirements, while recyclable waste feedstocks (such as cooking oils and yellow grease) have limited availability. Thus, interest in using sewage sludge as a feedstock for biodiesel production has grown significantly [49] due to its excellent economic viability and a promising future intensification in its research.

In Europe, the country with the highest annual production of sewage sludge is Germany (1.85 million tons of dry solids), followed by the UK (1.14 million tons), and Spain (1.03 million tons). It can also be mentioned that more sewage sludge is produced in Japan than in the member countries in the EU. Despite scarce data on the USA, it is believed that its sludge production is the second largest in the world. China, being the country with the largest population (about 1.3 billion), has been experiencing a considerable continuous increase in its sewage sludge generation since 2011, and it surpassed the 12-million-ton mark in 2017. In that year, 45 million tons of this waste were produced worldwide [50]. Therefore, due to the huge economic potential and growth of municipal sewage sludge production, this constitutes a raw material that may become a strong trend in the future in this area.

Since about 85% of the total biodiesel production value is linked to feedstock, lipid-rich sewage sludge is also highly interesting for biodiesel production from an economic standpoint [51]. As stated in the European Union Report (Commission, 2008b), about 10 million tons of dry solids of sewage sludge were produced in the 26 European Union Member States in 2008, of which 36% (or 3.7 million tons of dry solids) was recycled for use in agriculture [52]. This shows that the material has great potential for production increase, given the urbanization rates seen worldwide.

Sewage sludge comprises organic materials (lipids, proteins, carbohydrates, etc.) and inorganic materials (heavy metals and ash, for example) from alternative waste treatment processes. It is an unavoidable waste, its production is constantly increasing, and it causes great concern regarding the risks imposed to human health and its significant potential for environmental damage. For this reason, strict regulations have limited the disposal of sewage sludge, causing alternative methods to be sought, including technological processes that employ principles of thermochemistry, such as gasification, pyrolysis, and direct combustion [53]. Using sludge for biodiesel production is an ecologically positive methodology for reusing and recycling this waste in the obtainment of versatile products via fermentation technology, with the potential aid of microorganisms. These approaches can significantly curb the amount of sludge disposal and provide high-value-added products with a lower cost of manufacture [54].

Figure 11A,B shows the number and ratio of occurrences among some of the raw materials used in the production of biodiesel and their presence in the titles of the 4586 analyzed documents. By analyzing this graph, microalgae and cooking oil residue, as previously mentioned, occupy a prominent place among the documents. Sewage sludge is the feedstock with the lowest relevance among these, as this is still in the research development stage and may be an emerging trend in the future of the scientific literature in this area. Biodiesel production using sewage sludge as a feedstock is surrounded by complex challenges that must be addressed before it can see a breakthrough in the market. Some of these challenges are not unique to biodiesel production, but to the biofuel industry in general. A few of these challenges include sludge waste collection, product quality management, regulatory concerns, suboptimal yields, process economics, pharmaceutical and chemical contaminants, soap formation, and product separation, among others [35].

Other advantages of sewage sludge are the fact that it is abundant and can deliver a continuous yield. The higher the value of food commodities in the future, the more competitive sewage sludge is set to become as a biodiesel feedstock [55]. Considering the facts discussed so far, it is clear that a future viability of sewage sludge is highly probable, given the drive towards ecological change for global sustainability. Municipal sewage sludge has a high potential to be a reliable and high-energy feedstock for future use in biodiesel production [56]. Increasingly more studies have been carried out on this topic, as shown in the graph related to Figure 1, which points to a sustainable interest in more intense biodiesel production using innovative feedstocks that contribute to this purpose.

Figure 12 illustrates the occurrence of the keywords from the 60 articles collected, and it can be seen that other related terms are also found, such as “lipid extraction” and “transesterification method”, which are topics highly discussed in the content of the above documents. In addition, it is possible to notice the presence of discussions around microalgae that, although not the main focus of the articles in question, is clearly seen as a raw material of extreme importance in the literature on biofuel production. Figure 12 is a quantitative representation in the form of a density map, which reveals the prominence of some keywords over others, and enables the observation of a hierarchical formation of the organization of the keywords according to the number of occurrences.

#### 3.2.3. Microalgae

As discussed earlier in this article, microalgae are an extremely important raw material in most of the articles analyzed, given the fact they are the main subject of articles with very large numbers of citations. Again, the most cited article is entitled Microalgae for biodiesel production and other applications: A review [27], which has accumulated 3160 citations. Microalgae are composed of many photosynthetic microorganisms that can use carbon dioxide from the atmosphere to produce biomass faster and more efficiently than land plants. They are considered promising raw materials in various industry segments, such as feed, biofuels, food, pharmaceuticals, and nutraceuticals [57]. It is estimated that there are more than 100,000 species of algae. These plants can be used as biofertilizers for aquaculture, animal feed, and food, among other applications [58].

Microalgae cultivation can be carried out in environments considered unsuitable for the cultivation of other plants, such as in saltwater, brackish water, fresh water, or non-arable land that is unsuitable for conventional farming methods. In addition, cultivation can also be done in farms or even in bioreactors. Due to their non-selective, per hectare growth and development, microalgae can deliver higher production yields with better ecological performance [34]. Harvesting microalgae can cost as little as 20% of the total biodiesel production cost, depending on the farming method used [59].

Three main problems in algae-based biofuel production have been highlighted: supporting algae cultivation in different climates, the high water demand, and the technology deficit for commercialization. Nevertheless, the market is predicted to create future local and regional partnerships and collaborations with a view to maximize adoption and enable technologies for manufacturing them at a larger scale [60]. We can then conclude that this continues to be a raw material of extreme importance in the studies, and for the future, of the biofuel industry.

The potential of microalgae through high lipid productivity using small land areas, as well as the ability to use unproductive lands to this end, justify the recent investments in their culturing for fuels [61]. Therefore, given that there are significant economic advantages to the cultivation of this raw material and that the studies involving it are on the rise, microalgae are strong players among the raw materials currently used for biodiesel production, especially from a sustainability standpoint.

#### 3.2.4. Waste Cooking Oil

After performing occurrence and relevance analyses, we also identify waste cooking oil as a feedstock of high interest in the research on biofuel production. Among the documents collected on this area from 2010 to 2021, waste cooking oil occupies the second place in the ranking of occurrences of articles with this specific feedstock in their title, with 221 documents (4.8% of the total). Moreover, according to Figure 8, which shows the keywords for the analysis of the emerging future trends in the research related to biodiesel production in 2020 and 2021, used cooking oil is repeatedly shown even among other terms related to methods and reactions inherent in different research topics (such as “transesterification”, “heterogeneous catalyst”, and “conversion”).

Waste cooking oil is obtained from edible oils that have been used for frying food [62]. These types of food wastes are harmful to human health and to the environment when they are improperly disposed of by not submitting them to any treatment processes [63]. Waste cooking oil is seen as a promising feedstock for biodiesel production due to its low cost and abundance in several countries [64]. The employment of previously used cooking oils as raw materials does not create controversial issues, such as discussions about the clash between the food and fuel industries, or environmental reservations [63]. The use of this raw material for the production of biofuels was first reported a long time ago, and research on the topic started as early as the 1970s [65]. Waste cooking oil can be obtained from homes, hotels, restaurants, and food businesses that utilize frying operations and other similar food preparation processes [66]. Waste oils generated from household, commercial, and industrial sectors can be easily converted into biodiesel [67]. The production of biodiesel using used cooking oil is environmentally friendly and is a recognized solution in waste management practices [68].

Besides these raw materials, several others can be highlighted, especially in the non-edible group of materials. An example is beef tallow, which in Brazil is the second main raw material used in the production of biofuels after soybean oil. Biodiesel that is produced from tallow only generates between 17% and 35% of the total impact caused by low-sulfur diesel, thus showing an environmental advantage of using the former [69]. Another material is J. hieronymi, an endemic species in the semi-arid and arid northwestern region of Argentina, whose oil is not edible. It is a non-conventional oilseed species that does not compete with the food industry and hence has great economic potential as an alternative oil [70].

This work reveals several aspects of the biodiesel research that are clearly present across previously published review articles, such as the interest in researching feedstocks for biodiesel production in Asia, in view of the needs and economic specificities of this region. Additionally, new emerging trends are evidenced from the subsequent analysis of the articles published within the years delimited in this research, which also shows a constant search for sustainability approaches that are coupled with the maintenance of economic viability. These research trends are directed towards raw materials that do not create market competition with other industries, which is the case of the food sector, which in many cases utilizes a range of raw materials that can also be used for biodiesel production. Feedstocks such as sewage sludge, waste oils, and non-edible vegetable oils are the research foci in many of the papers reviewed earlier in this work.

## 4. Trendy Research Topics

### 4.1. Quantitative Analysis of Frequent Keywords

To better understand the development of the research on a specific area, it is necessary to carry out an analysis of the keywords in the published documents, as they reveal essential information about the topic, such as applications, trends, relevance of documents, discussions, and other general research characteristics. Table 4 lists the 24 most prominent keywords mentioned in the articles analyzed. The top six keywords are biodiesel (225), transesterification (145), residual cooking oil (91), optimization (62), soybean raw materials (60), and esterification (50).

Figure 13 illustrates the visualization network map generated from the VOSviewer program of the 60 most important keywords with at least eight occurrences in the collated database. The “biodiesel” keyword is clearly the largest group, belonging to the yellow cluster. This keyword is linked to “feedstocks”, “microalgae,” and “oil”, which define the raw materials used in the research and are interconnected. The keyword “transesterification” is the second most reported. It is linked to “waste cooking oil” and “vegetable oil”, which are also part of the blue cluster. Transesterification is the procedure that replaces the organic alkyl groups of a vegetable/plant oil (an ester) with a methyl alcohol group [71]. The cluster of optimization and general optimization encompasses some highly relevant topics in the search, such as emission characteristics, heterogeneous optimization, process optimization, and engine performance, giving us a good view into the important elements in the field of optimization. Regarding the methodologies and materials, it can be seen that this specific cluster illustrates the relationship of some keywords with “conversion”, “esterification”, “methane”, and “acid”. 

### 4.2. Research Areas

The CiteSpace software was used to organize the collated database and to analyze the emerging trends in the area. As mentioned earlier in this article, CiteSpace allows the visualization of knowledge development in a given area of research [72]. Analyzing a cluster enables the identification of study themes and data of greater relevance to the specific research [73]. Keywords can determine the future research paths related to biodiesel production and its raw materials. Table 5 illustrates the six primary sets of co-citations among the articles linked to this topic. 

#### 4.2.1. Research Fields

Cluster #0 has “oleaginous yeast” as its main keyword. These yeasts have gained significant prominence worldwide in metabolic engineering due to the fact they offer facilitated pathway manipulation, fatty acid- or oleochemical-derived metabolite enhancements, and simplified cultivation strategies [74]. The article that represents the cluster was published by Chtourou [75], and the main objective of their research was to perform the investigation of lipid accumulation in, and growth of, a new isolated marine microalgae strain. The optimization of the composition of the culture medium and the application of different stressful environmental conditions were used to this end. El-Sheekh [76] performed research on different species of isolated Scenedesmus, comparing their efficiency as feedstock for biodiesel production. The third reference article of the cluster was written by Abomohra [77], in which ten macroalgae were collected and selected as a biodiesel feedstock. The research confirmed that macroalgae are desirable potential alternatives as renewable feedstocks for biodiesel production.

Cluster #1 is represented by the keyword “ionic liquid”. Ionic liquids are chemical compounds with one cation and one anion, defined by melting points of below 100 °C. Each of the above ions allows for the insertion of a unique property or function into a molecule [78]. Mansir [79] conducted research whose main objective was to demonstrate the current status of using heterogeneous bifunctional acid/base catalysts for biodiesel production from green and non-edible waste cooking oil. Zhang [80] was responsible for conducting a study that proposed a practical method used for biodiesel production from high FFA feedstocks with a high reaction rate, fewer environmental problems, less toxicity, and minimal corrosion. Petchsoongsakul [81], who published an article that showed great prominence in cluster #1, presented a novel hybridization of transesterification and esterification processes in a single reactive distillation column to be used in the production of biodiesel from waste cooking oil.

#### 4.2.2. Emerging Trends

Cluster #2 shows the process designs involved in the various methodologies for producing biodiesel from feedstocks. One work [82] reviewed the transesterification process of low-content feedstocks for conversion to biodiesel via supercritical fluid technology, which is an environmentally friendly technique that shows higher process efficiency. Soares [83], in turn, performed an investigation of a new strategy used for a hydroesterification-based biodiesel production from low-cost oil feedstocks. This involves the complete hydrolysis of the feedstock to fatty acids in subcritical water, followed by the use of a packed bed reactor, which contains a fermented solid with lipase that performs the conversion of fatty acids into their ethyl esters. Jain [84] conducted a review of the kinetics of biodiesel production and reveals the results obtained from a two-step kinetic study of the acid–base-catalyzed transesterification process performed at preset temperatures of 65 and 50 °C in the esterification and transesterification process, respectively, under an optimal methanol-to-oil condition.

Cluster #3 depicts oil extraction processes, listing the studies covering different methodologies to this end. [85] carried out a study that presented, compared, and discussed several potential feedstocks for biofuel production, along with several oil extraction methods, and the advantages and disadvantages of using the different methodologies. [86], the second most relevant paper in cluster #3, deals with the biodiesel production from non-edible seeds, specifically by using *Hevea brasiliensis* (HB) and *Ricinus communis* (RC) as potential feedstocks. An esterification–neutralization–transesterification (ENT) process was used for biodiesel production. [87] investigates the potential of the promising feedstock *Calophyllum inophyllumas* for biodiesel production. The author assessed many crucial aspects of this process, such as the chemical and physical properties of the *Calophyllum inophyllum* crude oil and methyl ester, the mixture and engine performances, fatty acid composition, and the emissions of the *Calophyllum inophyllum* methyl ester.

In cluster #4, the focus is on the production of enzymatic biodiesel. Adachi [88] attempted to carry out the integration of a lipase-catalyzed ethanolysis and a fermentative bioethanol production process. Lokman’s work [89], on the other hand, covers and explores the joint use of feedstocks that are considered to be of lower quality in tandem with carbon-based catalysts to perform the conversion of a refinery crude palm oil residue that contains a high percentage of free fatty acids. The production and characterization of the carbohydrate-derived solid acid catalysts were critically discussed, also with a view to measure their physicochemical properties. Another key article in this cluster is that of Aransiola [38], which reviews the various technologies used for biodiesel production to date, with a view to compare the commercial conformation of these methods based on feedstock availability. It was noted that there is a strong emphasis on using microalgae oil sources. The economic viability of the process is still a point that needs further discussion.

In cluster #5, calcium oxide is emphasized. CaO is an inexpensive material that can be found in abundant quantities in the Earth’s lower mantle and a material that does not cause harm to the environment [90]. The prominent paper by Mazaheri [91] shows an overview of the advances in the use of calcium oxide-based catalysts in biodiesel production. The paper highlights the various factors involved in the synthesis of calcium oxide-based catalysts, and, furthermore, the effect of reaction parameters on their yield for biodiesel production is assessed. Another central article produced by Penarrubia [92] shows an industrial-scale simulation performed to compare the traditional alkali-catalyzed process via sodium hydroxide catalysis to an enzyme-catalyzed process that was developed by the research group of one of the authors involved. Finally, Moser’s work [93] investigates the fuel characteristics that are highly dependent on the fatty acid composition of the feedstocks used in biodiesel production. Thus, the fatty acid profile was defined as a powerful screening tool to select feedstocks rich in monounsaturated fatty acids for further evaluation.

#### 4.2.3. Two Key Insights

One issue with systematic reviews is that they may present data and conclusions within a confidence level that does not always reflect the reality, or that is feasible. One example is when systematic reviews for a particular emerging field of study do not yet exist. In addition, reviews that have been conducted within a certain time period may be out of date or may not have included all the scientific advances as comprehensively as it should. The CiteSpace tool was created to provide a potential solution to these challenges by enabling the use of customized datasets to answer questions about a field of knowledge that is changing rapidly [94]. It can also help extract valuable information from articles published on a specific subject matter. However, there is a need to consider its limitations concerning the optimization tools and the accuracy of the analysis of information from the database.

Node size is one of the key analysis points verified by CiteSpace. In Table 5, the data regarding the node size of each cluster in the collected database are measured, thus allowing for the quantification of the relevance of each cluster. It is also possible to observe that, among those listed, the cluster with the most recent node size (2016) is the one in position #5 and has “calcium oxide” as its most prominent keyword.

The first perspective pertains to studies related to the feedstocks used for biodiesel production. The selection of raw materials is a significant point dictating the quality and cost of the biodiesel produced [95]. There is some reported competition between the biodiesel and the edible oil markets, as many feedstocks are of use to both. Many studies have recently been performed with a view to try to find alternative feedstocks from non-edible sources that are low-cost and sustainable [44]. Performing a shelf-life analysis of feedstocks is crucial in biodiesel production [96], and many important aspects define the assessments made in the research related to these feedstocks. Attention to environmental degradation is becoming increasingly more important, and the scientific community sees the urgent need to improve the production process of renewable alternatives to petroleum-based fuels, and also to develop new ones. There is a high possibility that renewable fuels are to become an essential product in bio-based economies [97].

The second important insight from this work regards the production of biodiesel from sewage sludge, which can certainly provide a more sustainable angle to such processes, given the environmental need for this. Using sewage sludge as a feedstock for biodiesel production and therefore offering options to the resolution of the food–fuel debate can also help solve some of the difficulties in treating sludge [49]. Biodiesel from sludge can be a significantly superior alternative to other materials used as food-based feedstocks in biodiesel production. As mentioned earlier, the material has very low or zero added cost and is sustainably generated in wastewater treatment plants. The production value of the resulting biodiesel would be much lower when compared to that produced from other sources. This would eliminate the cost of feedstock materials, which is the most expensive element in biodiesel production and can account for almost 90% of the total cost of manufacture [53].

## 5. Conclusions

This article provided a comprehensive view into the literature related to feedstocks for biodiesel production, the emerging trends that are the focus of current work, and the promising alternatives to be explored in the future. In this work, a search methodology was devised to refine the search for, and the selection of, a set of articles, which started with a database of 4586 documents. From this initial database, two more targeted databases of 330 and 60 articles were created, respectively, in order to assess the direction of travel in this research area. The period analyzed (2010 to 2021) reveals a particular emergence of meaningful research relationships and collaborations, demonstrating a more sustainable vision among researchers for developing their work. Important conclusions from this review include:One of the key raw materials highlighted in this work, sewage sludge, has received little attention in the literature, considering that the number of cited articles is still small and despite the economic and sustainability advantages cited above. This is mostly due to the fact it is still a recent research theme and that raw materials that have been explored for longer are more attractive owing to the vast knowledge repertoire.Waste cooking oil and microalgae are the raw materials of most significant presence in the academic outputs analyzed. These are feedstocks that have been extensively reported on in the literature, mainly due to their long-standing reputation in the area. However, there has been a noticeable reduction in the volume of cited articles over the years.China, Malaysia, and India are the countries with the greatest research outputs relating to feedstocks for biodiesel production. It can be concluded, therefore, that Asia shows a great interest in this area. One factor that can explain this interest may be the sheer number of people concentrated in this region, since China, India, and Malaysia account for more than 35% of the world’s total population.For a more specific analysis of this research, keywords collected by the CiteSpace program were used. From a systematic verification of the terms, it was possible to observe that the main research topics in this area include oleaginous yeasts, ionic liquids, process design, oil extraction, enzymatic biodiesel production, and calcium oxide.Two broad perspectives related to this research area have been emphasized. The first is the generalized view of the articles that engage in the topic of feedstocks for biodiesel production. It is concluded that there is a great deal of discussion regarding the economy versus sustainability dilemma, and researchers have sought practical solutions to the problems that arise from this. The second perspective concerns the recent emergence of academic interest in studying sewage sludge for biodiesel production. It is understood that this area will be further explored in the near future due to the solution that this raw material represents to the conundrum above.

Future development prospects in the biofuel market are mostly linked to a vision of sustainable change, as it was possible to observe with the analysis of the growth in publication numbers on the topic. Importantly, there is a clear concern from the countries that are active contributors to the global environmental degradation observed over the years to seek to reduce the impacts caused by them. The most influential researchers in this area are those who seek to fly the sustainability flag, as exemplified by Ahmad’s article and also by the research topics underlying hundreds of papers in our database. In addition, a great economic interest and concern was observed in most of the above works, where authors described processes on the basis of financial costs and time or productivity. This was the case for works that had sewage sludge as their main theme, for example, which categorically sought to explain the advantages and productivity hurdles of this raw material and ways in which it could become a key trend within the biodiesel market.

The direction of the literature concerning the research topic under study is constantly being shaped due to the worldwide need for sustainability and a more environmentally centered economy. A very pertinent problem is the alleged competition between the biodiesel and the food industries for edible raw materials, in which, due to scale reasons, the latter industry will always have a competitive advantage over the former. The solution to this is in the exploitation of raw materials that are derived from non-edible oils. The environmental concern of researchers has shaped the methodological advances for using these oils in a way that can reconcile sustainability with the cost of manufacture, given that, even if a feedstock is exceptionally sustainable, it can be economically unviable.

The research contained in this review proposed to elucidate and explore the directions of travel of the research that will define the future approach to biodiesel production, highlighting the concerns regarding environmental degradation. Nevertheless, there are some limitations in this work, a major one being the fact that the Web of Science was the only database used to create our own databases. Apart from the refinements made (language and type of documents), other articles were certainly inadvertently excluded by not being in this specific database.

Furthermore, it is necessary to emphasize that the evaluation methodology used in this work was developed from the individual vision of the research described throughout the text. This should be considered, since several ways of approaching this research area have led to significantly different results. It could also be highlighted that the research domain encompassing feedstocks for biodiesel production is highly diverse due to the wide variety of research approaches undertaken on the topic. There is a clear evolving trend towards linking academic work and social compliance for sustainability. Some aspects of collaborative work were verified, showing that some countries still opt for internalized cooperation in their research development, in which authors mostly collaborate with others of the same nationality. The scientific literature on feedstocks for biodiesel production is extremely vast and can be analyzed further to allow for a comprehensive understanding of its specific characteristics that are of interest to researchers, driven mainly by a pressing, worldwide need for studies in this area.

## Figures and Tables

**Figure 1 bioengineering-09-00539-f001:**
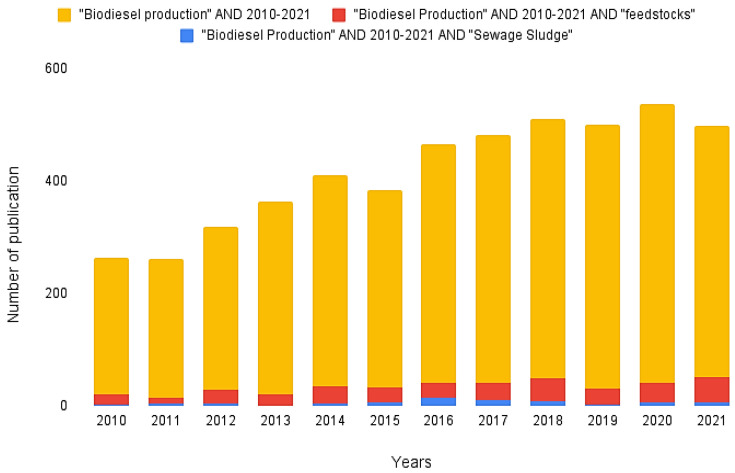
Annual publications on biodiesel production, relevant raw materials, and on sewage sludge as a resource. The Google Sheets tool was used to produce this figure.

**Figure 2 bioengineering-09-00539-f002:**
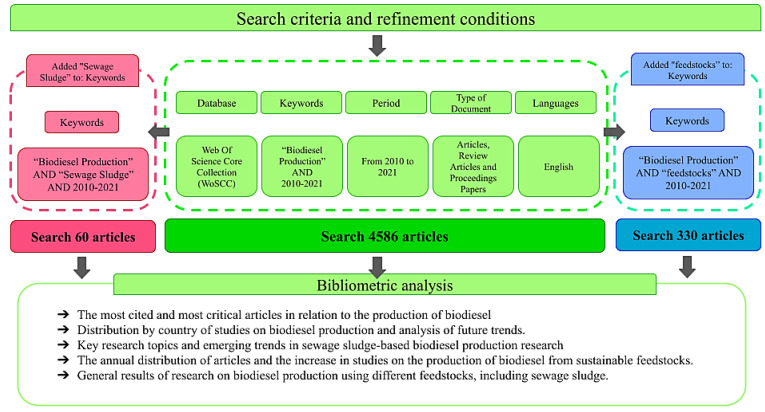
Search and analysis criteria used for the data collection stage.

**Figure 3 bioengineering-09-00539-f003:**
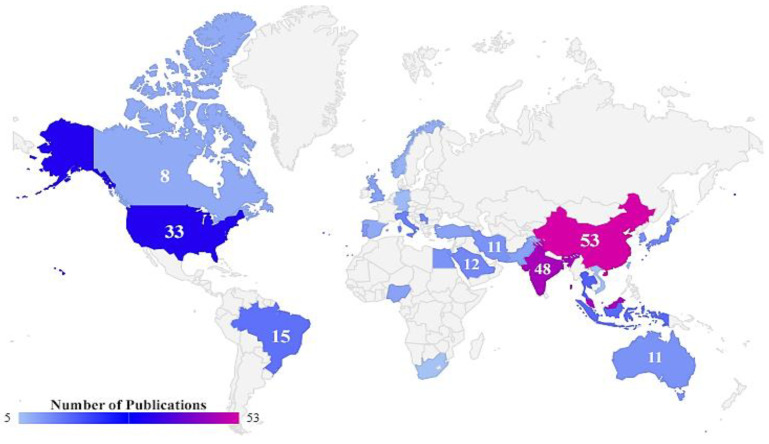
Representation of the distribution of articles by country.

**Figure 4 bioengineering-09-00539-f004:**
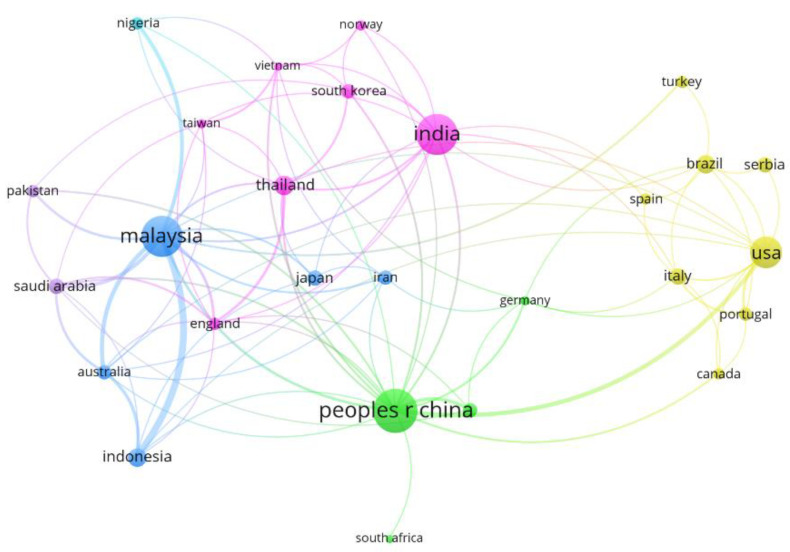
Network visualization map showing the collaboration among countries with at least five published papers. The thickness of the lines connecting two countries indicates the accumulation of co-authorships (thicker lines means more published articles), and the color clusters illustrate the groups of countries with a high level of collaboration.

**Figure 5 bioengineering-09-00539-f005:**
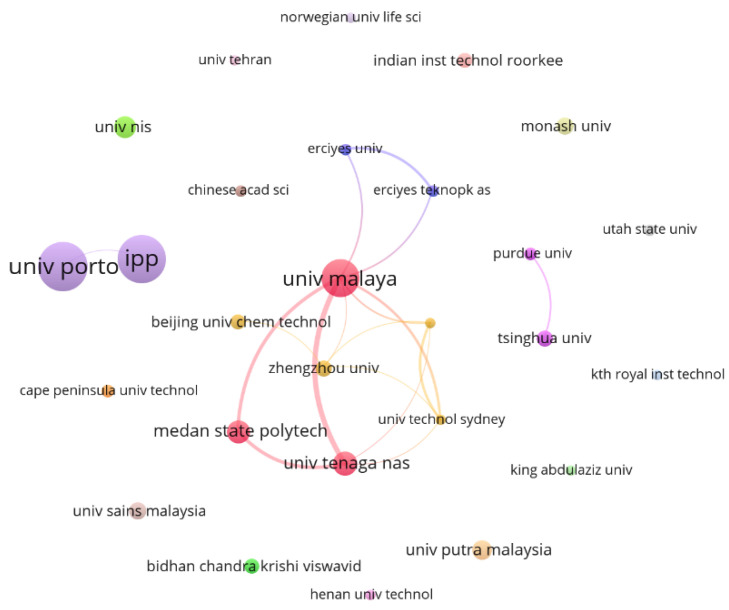
Network visualization map showing the collaboration among organizations with at least 250 accumulated citations. The thickness of the lines connecting two organizations indicates the accumulation of co-authorships (thicker lines means more joint published articles), and the color clusters highlight the groups of institutions with a high level of collaboration.

**Figure 6 bioengineering-09-00539-f006:**
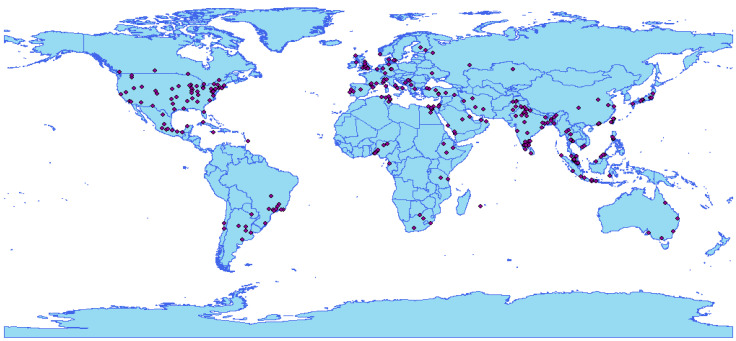
Geocoding of the organizations responsible for the publication of the 332 articles analyzed.

**Figure 7 bioengineering-09-00539-f007:**
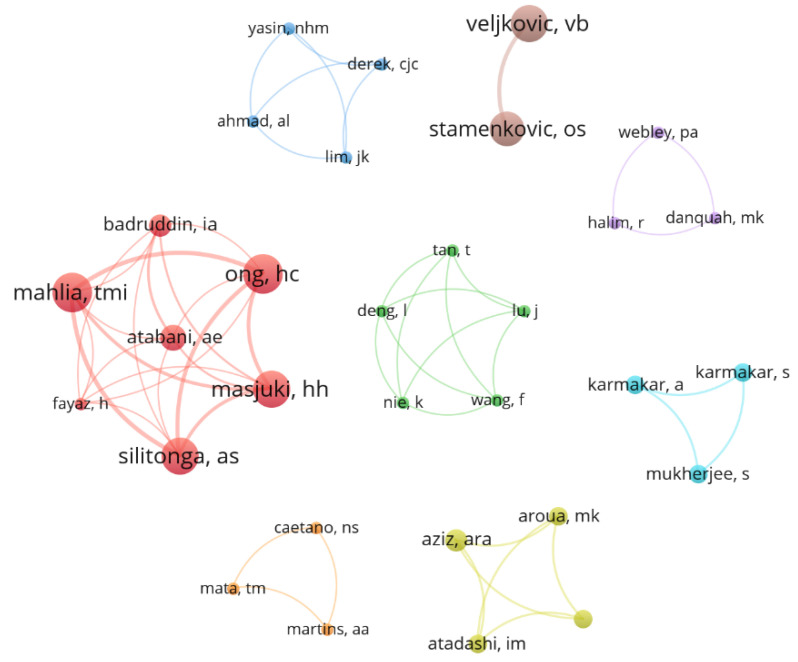
Network map showing the collaboration among authors with at least four publications. The thickness of the lines connecting two authors indicates the accumulation of co-authorships (thicker lines means more published articles), and the color clusters illustrate the groups of authors with a high level of collaboration.

**Figure 8 bioengineering-09-00539-f008:**
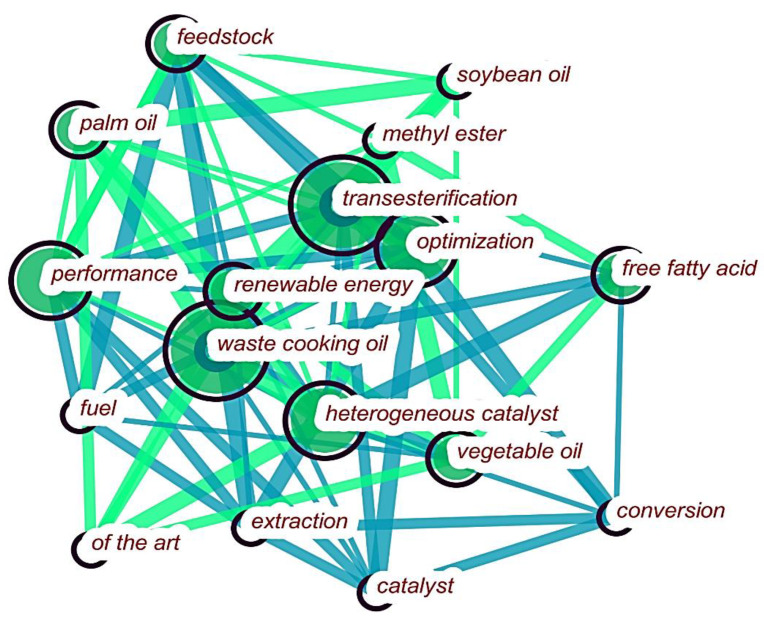
Keyword visualization network (January 2020 to December 2021).

**Figure 9 bioengineering-09-00539-f009:**
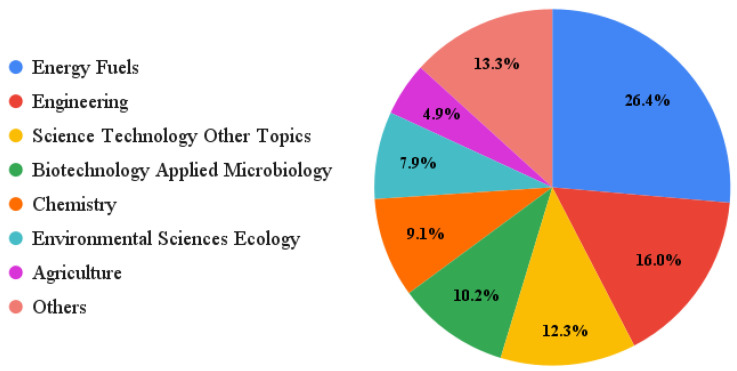
Distribution of research areas related to feedstock for biodiesel production.

**Figure 10 bioengineering-09-00539-f010:**
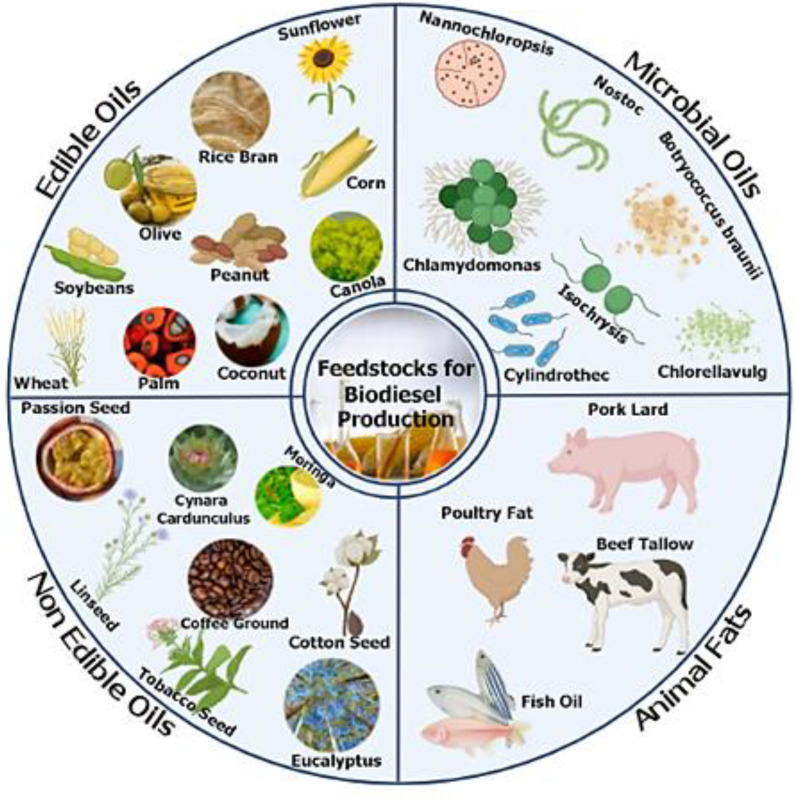
Various feedstocks used in the production of biodiesel. These are divided into four classes: edible oils, inedible oils, microbial oils, and animal fats.

**Figure 11 bioengineering-09-00539-f011:**
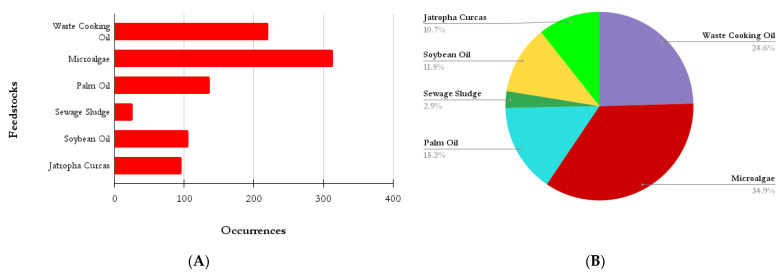
(**A**) Number of occurrences mentioning different feedstocks in the titles of their respective articles. (**B**) Ratio showing the percentage of articles on specific feedstocks that mention them in their respective titles.

**Figure 12 bioengineering-09-00539-f012:**
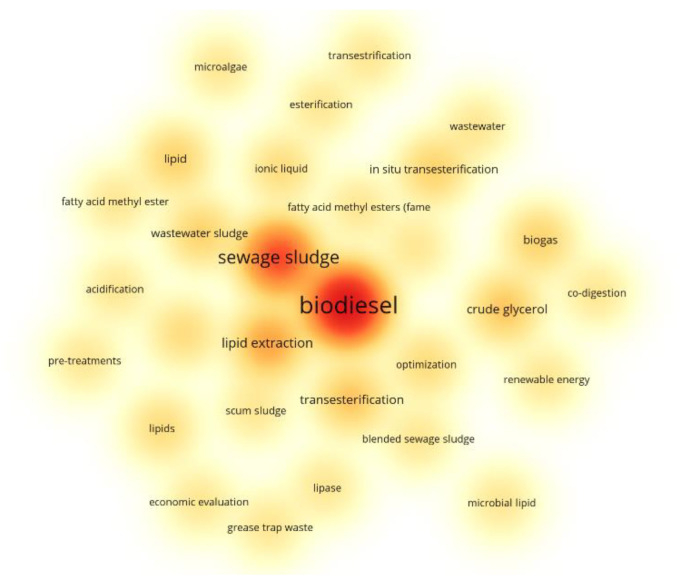
Density map of the database keywords relating to biodiesel production from sewage sludge. More intense color points indicate a higher number of keyword occurrences.

**Figure 13 bioengineering-09-00539-f013:**
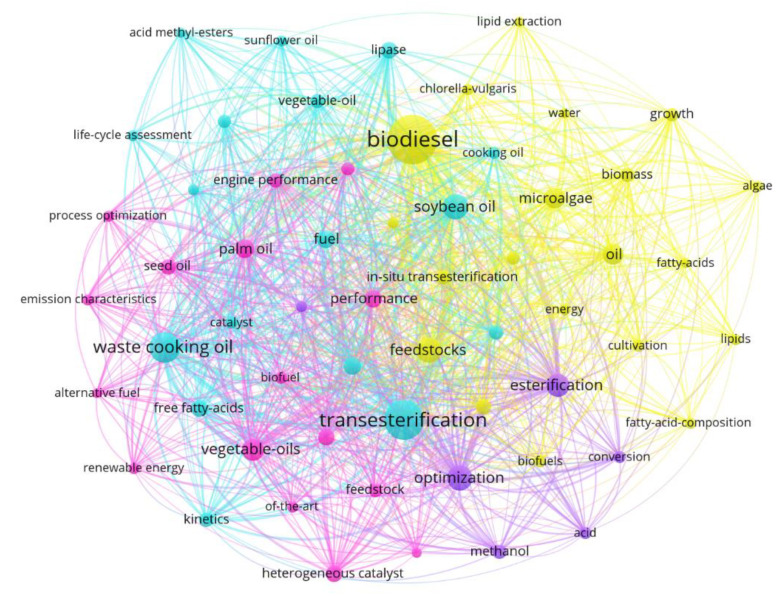
Visualization map of the co-citation network of keywords in the research related to raw materials for biodiesel production.

**Table 1 bioengineering-09-00539-t001:** Top ten scientific journals with publications in the field of feedstocks for biodiesel production.

Rank	Journal	C	IF	NP	NC	AC	P
1	Renewable & sustainable energy reviews	USA	14.982	33	8250	250	9.9%
2	Bioresource technology	NL	9.642	16	1785	111.5	4.8%
3	Energy conversion and management	EN	9.709	13	789	61.4	3.9%
4	Biomass & bioenergy	EN	5.061	6	750	125	1.8%
5	Fuel	EN	6.609	18	729	40.5	5.4%
6	Biofuels, bioproducts & biorefining—Biofpr	EN	4.102	5	151	30.2	1.5%
7	Journal of environmental chemical engineering	EN	5.909	4	63	15.7	1.2%
8	Catalysts	CH	4.146	6	59	9.8	1.8%
9	Biofuels-UK	EN	2.956	4	57	14.2	1.2%
10	Environmental chemistry letters	GER	9.027	4	53	13.2	1.2%

C = Country; IF = Impact Factor in 2020; NP = Number of Publications; NC = Number of Citations; AC = Average Citations; P = Percentage in Relation to the Total Number of Papers. USA = United States of America; NL = Netherlands; EN = England; CH = Switzerland; GER = Germany.

**Table 2 bioengineering-09-00539-t002:** The 10 most prolific countries in the area of feedstocks for biodiesel production.

Rank	Country	NP	NC	AC	Total Link Strength	AC
1	China	53	2555	48.2	169	250
2	Malaysia	48	3715	77.4	353	111.5
3	India	48	2190	45.6	252	61.4
4	United States of America	33	2030	61.5	116	125
5	Thailand	17	164	9.6	55	40.5
6	Indonesia	16	1309	81.8	134	30.2
7	Brazil	15	531	35.4	51	15.7
8	Italy	13	164	12.6	42	9.8
9	Serbia	12	920	76.6	82	14.2
10	Saudi Arabia	12	492	41.0	100	13.2

NP = Number of Publications; NC = Number of citations; AC = Average citations (NC/NP).

**Table 3 bioengineering-09-00539-t003:** Most cited papers in the field of feedstocks for biodiesel production.

Rank	Article Title	Authors	Year Published	Citations
1	Microalgae for biodiesel production and other applications: A review [27].	Mata, Teresa M.; Martins, Antonio A.; Caetano, Nidia. S.	2010	3162
2	Non-edible vegetable oils: A critical evaluation of oil extraction, fatty acid compositions, biodiesel production, characteristics, engine performance, and emissions production [33].	Atabani, A. E.; Silitonga, A. S.; Ong, H. C.; Mahlia, T. M. I.; Masjuki, H. H.; Badruddin, Irfan Anjum; Fayaz, H.	2013	645
3	Microalgae as a sustainable energy source for biodiesel production: A review [34].	Ahmad, A. L.; Yasin, N. H. Mat; Derek, C. J. C.; Lim, J. K.	2011	590
4	Extraction of oil from microalgae for biodiesel production: A review [35].	Halim, Ronald; Danquah, Michael K.; Webley, Paul A.	2012	586
5	Biodiesel production with immobilized lipase: A review [36].	Tan, Tianwei; Lu, Jike; Nie, Kaili; Deng, Li; Wang, Fang	2010	457
6	Properties of various plants and animal feedstocks for biodiesel production [37].	Karmakar, Aninidita; Karmakar, Subrata; Mukherjee, Souti	2010	421
7	A review of current technology for biodiesel production: State of the art [38].	Aransiola, E. F.; Ojumu, T. V.; Oyekola, O. O.; Madzimbamuto, T. F.; Ikhu-Omoregbe, D. I. O.	2014	368
8	A review of biodiesel production from jatropha curcas L. oil [39].	Koh, May Ying; Ghazi, Tinia Idaty Mohd	2011	365
9	Biodiesel production by simultaneous extraction and conversion of total lipids from microalgae, cyanobacteria, and wild mixed-cultures [40].	Wahlen, Bradley D.; Willis, Robert M.; Seefeldt, Lance C.	2011	319
10	The effects of catalysts in biodiesel production: A review [41].	Atadashi, I. M.; Aroua, M. K.; Aziz, A. R. Abdul; Sulaiman, N. M. N.	2013	308

**Table 4 bioengineering-09-00539-t004:** Ranking of the 24 most prominent keywords mentioned in the analyzed articles.

Rank	Keyword	Frequency	TLS	Rank	Keyword	Frequency	TLS
1	biodiesel	225	978	13	performance	31	170
2	transesterification	145	714	14	fuel production	31	183
3	waste cooking oil	91	475	15	lipase	27	136
4	optimization	62	323	16	*Jatropha curcas*	26	130
5	feedstocks	61	285	17	heterogeneous catalyst	26	137
6	soybean oil	60	336	18	free fatty acids	25	158
7	esterification	50	272	19	seed oil	24	133
8	vegetable oil	47	252	20	extraction	22	100
9	oil	40	149	21	in situ transesterification	20	104
10	microalgae	40	189	22	rapeseed oil	20	111
11	palm oil	35	205	23	vegetable oil	20	111
12	fuel	31	159	24	kinetics	20	127

**Note:** TLS: Total Link Strength.

**Table 5 bioengineering-09-00539-t005:** Top six co-citation research clusters on raw materials for biodiesel production based on the CiteSpace analysis.

CID	Label	NS	Mean	Top Five Terms	Representative Articles
#0	oleaginous yeast	70	2015	oleaginous yeast; lipid extraction; wild strain; Egyptian freshwater habitat; different species.	(CHTOUROU, 2015) and (EL-SHEEKH, 2018)
#1	ionic liquid	63	2014	ionic liquid; ultrasound-assisted transesterification; using calcium oxide catalyst; economic variable; acyl acceptor.	(ZHANG, 2010) and (MANSIR, 2018)
#2	process design	54	2013	process design; heterogeneous catalyst; subcritical water; packed bed reactor; soybean soap stock acid oil.	(ZENG, 2014) and (SOARES, 2013)
#3	oil extraction	47	2014	oil extraction; plant seed; *Ricinus communis*; *Hevea brasiliensis*; *Calophyllum inophyllum L.*	(KENENI, 2017) and (SILITONGA, 2016)
#4	enzymatic biodiesel production	42	2013	enzymatic biodiesel production; low-cost feedstock; carbohydrate-derived solid acid catalyst; brown rice; ethanol fermentation.	(ADACHI, 2016) and (LOKMAN, 2014)
#5	calcium oxide	27	2016	calcium oxide; current state; technological progress; different type; LCA studies.	(MAZAHERI, 2021) and (FERNANDEZ, PENARRUBIA, 2017)

**Note:** CID = Cluster ID, NS = Node Size.

## Data Availability

The data presented in this study are available on request from the corresponding author.

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
