# Peer review of "Sustainable Feedstocks and Challenges in Biodiesel Production: An Advanced Bibliometric Analysis"

_bioengineering, 2022, doi:10.3390/bioengineering9100539_

Round 1
Reviewer 1 Report
The scope of the work “The feedstocks for sustainability and challenges in biodiesel production: An advanced bibliometric analysis” is relevant for the growth of the biodiesel sector and its research area. However, the following should be considered to enrich the manuscript.
Title: OK
Abstract:
the major findings or results from the analysis are missing in the abstract. It is worth highlighting some of the major findings/results and implications of the analysis in the abstract, authors should consider
Introduction:
The recent state of the art which this study results from is not shown in the introduction. A brief review of the literature showing the knowledge gaps identified and the link to the paper’s goals should be considered.
Materials and Methods: OK
Results and Discussion: The findings of the current work need to be compared and discussed with the results of other researchers/methods/scope findings. The authors should consider a comparison between the forecasting results or empirical results with the literature.
Conclusion: The conclusion section appears to be a detailed summary of results/observations. Authors should consider concise and convincing statements on what was found to be novel or impactful based on the results.
Author Response
Dear Ms. Lauren Liang,
We want to thank the reviewers for their crucial and constructive criticism, which allowed us to improve the scientific quality of our work immensely. Please find below the answers to their comments and questions. All modifications have been marked in yellow in the revised version of the manuscript to facilitate verification.
We Include a CONFIRMATION OF LANGUAGE EDITING SERVICES letter from Dr Micael de Andrade Lima Lecturer in Food Innovation Natural Resources Institute University of Greenwich. All modifications have been marked in red in the revised version of the manuscript to facilitate verification.
Reviewer 1 Review Report (Round 1)
Comments and Suggestions for Authors
The scope of the work “The feedstocks for sustainability and challenges in biodiesel production: An advanced bibliometric analysis” is relevant for the growth of the biodiesel sector and its research area. However, the following should be considered to enrich the manuscript.
Title: OK
Abstract:
the major findings or results from the analysis are missing in the abstract. It is worth highlighting some of the major findings/results and implications of the analysis in the abstract, authors should consider
Answer: Thank you for your comments and contributions. All suggestions were incorporated into the new version of the manuscript. Changes are in the revised manuscript.
Introduction:
The recent state of the art which this study results from is not shown in the introduction. A brief review of the literature showing the knowledge gaps identified and the link to the paper’s goals should be considered.
Answer: Thank you for your comments and contributions. All suggestions were incorporated into the new version of the manuscript. More citations have been inserted that contribute to the enrichment of the approach.
Materials and Methods: OK
Results and Discussion: The findings of the current work need to be compared and discussed with the results of other researchers/methods/scope findings. The authors should consider a comparison between the forecasting results or empirical results with the literature.
Answer: Thank you for your comments and contributions. All suggestions were incorporated into the new version of the manuscript. Changes are in the revised manuscript.
Conclusion: The conclusion section appears to be a detailed summary of results/observations. Authors should consider concise and convincing statements on what was found to be novel or impactful based on the results.
Answer: Thank you for your comments and contributions. All suggestions were incorporated into the new version of the manuscript. Changes are in the revised manuscript.
Reviewer 2 Report
1) Since there is no separate section to present the selected bibliography studied, this should be covered in the introduction and not in the methodology. However, very few bibliographic references were presented in the introduction, only five.
2) All bibliographic references must appear in order within the work, including [6].
3) Section 2, "Materials and Methods" needs to be expanded and developed, as it looks now being much too laconic.
4) Authors jump from reference [16] directly to citation [26].
5) Then the citation of reference [33] is missing.
6) Then the citations [39-41] are missing.
7) From citation [57] it jump directly to citations [69-71]. It's a total mess of citations.
8) Figure 2 requires a broader discussion.
9) Explain and discuss the conclusion and then statements in quotation marks: "Additionally, the studies carried out on the topics above have been successful in demonstrating the addressing of environmental concerns through the steps that have been taken to make the biofuel market more sustainable." (final of the 3.1.1).
10) Figure 12 is unclear.
11) After table 4 appears a final table numbered 3 (again 3) although it should be 5.
12) Before the conclusions section, insert a "Discussion" section in which to show all the new aspects proposed by the work, to highlight everything that the work brings new compared to the already existing articles, and to compare everything that is new with the already known aspects from the works of specialty studied by the authors.
Also here you recapitulate the main ideas launched by the paper and discuss their usefulness.
Author Response
Dear Ms. Lauren Liang,
We want to thank the reviewers for their crucial and constructive criticism, which allowed us to improve the scientific quality of our work immensely. Please find below the answers to their comments and questions. All modifications have been marked in yellow in the revised version of the manuscript to facilitate verification.
We Include a CONFIRMATION OF LANGUAGE EDITING SERVICES letter from Dr Micael de Andrade Lima Lecturer in Food Innovation Natural Resources Institute University of Greenwich. All modifications have been marked in red in the revised version of the manuscript to facilitate verification.
Reviewer 2 Review Report (Round 2)
Comments and Suggestions for Authors
1) Since there is no separate section to present the selected bibliography studied, this should be covered in the introduction and not in the methodology. However, very few bibliographic references were presented in the introduction, only five.
Answer: Thank you for your comments and contributions. All suggestions were incorporated into the new version of the manuscript. More citations have been inserted that contribute to the enrichment of the approach.
2) All bibliographic references must appear in order within the work, including [6].
Answer: All references and citations have been reorganized for better viewing. Thank you for your comments and contributions. All suggestions were incorporated into the new version of the manuscript. Changes are in the revised manuscript.
3) Section 2, "Materials and Methods" needs to be expanded and developed, as it looks now being much too laconic.
Answer: Thank you for your comments and contributions. All suggestions were incorporated into the new version of the manuscript. Changes are in the revised manuscript.
4) Authors jump from reference [16] directly to citation [26].
Answer: The citations and references have been reorganized for better viewing. Thank you for your comments and contributions. All suggestions were incorporated into the new version of the manuscript. Changes are in the revised manuscript.
5) Then the citation of reference [33] is missing.
Answer: The citation are properly inserted in 3.2.2. Sewage Sludge. Thank you for your comments and contributions. All suggestions were incorporated into the new version of the manuscript. Changes are in the revised manuscript.
6) Then the citations [39-41] are missing.
Answer: The citations are properly inserted in 3.2.3 Microalgae. Thank you for your comments and contributions. All suggestions were incorporated into the new version of the manuscript. Changes are in the revised manuscript.
7) From citation [57] it jump directly to citations [69-71]. It's a total mess of citations.
Answer: All references and citations have been reorganized for better viewing. Thank you for your comments and contributions. All suggestions were incorporated into the new version of the manuscript. Changes are in the revised manuscript.
8) Figure 2 requires a broader discussion.
Answer: Thank you for your comments and contributions. All suggestions were incorporated into the new version of the manuscript. Changes are in the revised manuscript. .
9) Explain and discuss the conclusion and then statements in quotation marks: "Additionally, the studies carried out on the topics above have been successful in demonstrating the addressing of environmental concerns through the steps that have been taken to make the biofuel market more sustainable." (final of the 3.1.1).
Answer: Thank you for your comments and contributions. All suggestions were incorporated into the new version of the manuscript. Changes are in the revised manuscript. This happens due to the prominence of the articles that are present in the literary repertoire used in the research, having in mind, mainly, the number of citations and renown of the authors present in the construction of the papers. These factors will be described in subsequent topics in the paper.
10) Figure 12 is unclear.
Answer: Thank you for your comments and contributions. All suggestions were incorporated into the new version of the manuscript. Additional information has been added that more clearly explains the purpose of this figure. Changes are in the revised manuscript..
11) After table 4 appears a final table numbered 3 (again 3) although it should be 5.
Answer: Thank you for your comments and contributions. All suggestions were incorporated into the new version of the manuscript. Changes are in the revised manuscript.
12) Before the conclusions section, insert a "Discussion" section in which to show all the new aspects proposed by the work, to highlight everything that the work brings new compared to the already existing articles, and to compare everything that is new with the already known aspects from the works of specialty studied by the authors.
Also here you recapitulate the main ideas launched by the paper and discuss their usefulness.
Answer: Thank you for your comments and contributions. All suggestions were incorporated into the new version of the manuscript. Changes are in the revised manuscript. Topic 3 deals with several aspects that were proposed in the paper, mainly the trends and bets for the sustainable future of biodiesel production. For example, the topic "3.2. Feedstocks for biodiesel production" demonstrates an analysis made from the verification of the database, revealing non-comestible raw materials and viable sustainability that showed up with great prominence.
Reviewer 3 Report
This review article provided comprehensive literature on feedstocks for biodiesel production and the emerging trends for biodiesel production. This article is nicely written and acceptable for the Bioengineering journal after minor revision.
Comments:
1) The authors provided comprehensive details and literature analysis on biodiesel production using different feedstocks. However, the future perspective of biodiesel production research is necessary to highlight in the article.
2) Provide more literature and discussion on the non-edible feedstocks apart from sewage sludge, such as different types of non-edible biomass feedstocks, if available.
3) What was the Source for Fig. 1? And using which tool (for example, SciFinder or any other software) this data (Fig. 1) was generated. Provide the source detail and how this data was generated information in the figure caption.
4) The reference style is not consistent. Please check the style of the all the references, specifically the journal name abbriviation.
Author Response
Dear Ms. Lauren Liang,
We want to thank the reviewers for their crucial and constructive criticism, which allowed us to improve the scientific quality of our work immensely. Please find below the answers to their comments and questions. All modifications have been marked in yellow in the revised version of the manuscript to facilitate verification.
We Include a CONFIRMATION OF LANGUAGE EDITING SERVICES letter from Dr Micael de Andrade Lima Lecturer in Food Innovation Natural Resources Institute University of Greenwich. All modifications have been marked in red in the revised version of the manuscript to facilitate verification.
Reviewer 3 Review Report (Round 3)
Comments and Suggestions for Authors
This review article provided comprehensive literature on feedstocks for biodiesel production and the emerging trends for biodiesel production. This article is nicely written and acceptable for the Bioengineering journal after minor revision.
Comments:
1) The authors provided comprehensive details and literature analysis on biodiesel production using different feedstocks. However, the future perspective of biodiesel production research is necessary to highlight in the article.
Answer: Thank you for your comments and contributions. All suggestions were incorporated into the new version of the manuscript. Changes are in the revised manuscript.
2) Provide more literature and discussion on the non-edible feedstocks apart from sewage sludge, such as different types of non-edible biomass feedstocks, if available.
Answer: Thank you for your comments and contributions. All suggestions were incorporated into the new version of the manuscript. Changes are in the revised manuscript.
3) What was the Source for Fig. 1? And using which tool (for example, SciFinder or any other software) this data (Fig. 1) was generated. Provide the source detail and how this data was generated information in the figure caption.
Answer: Thank you for your comments and contributions. All suggestions were incorporated into the new version of the manuscript. Changes are in the revised manuscript.
4) The reference style is not consistent. Please check the style of the all the references, specifically the journal name abbriviation.
Answer: Thank you for your comments and contributions. All suggestions were incorporated into the new version of the manuscript. Changes are in the revised manuscript.
Round 2
Reviewer 1 Report
The new version needs no more changes.
Author Response
Dear,
We want to thank them for their crucial and constructive criticism, which allowed us to improve the scientific quality of our work immensely. All modifications have been marked in the revised version of the manuscript to facilitate verification.
Reviewer 2 Report
Attention must still be paid to the numbering of the tables.
Author Response
Dear,
We want to thank them for their crucial and constructive criticism, which allowed us to improve the scientific quality of our work immensely. All modifications have been marked in the revised version of the manuscript to facilitate verification.
Kind regards.